# Bolus Intravenous Procainamide in Patients with Frequent Ventricular Ectopics during Cardiac Magnetic Resonance Scanning: A Way to Ensure High Quality Imaging

**DOI:** 10.3390/diagnostics11020178

**Published:** 2021-01-27

**Authors:** Chrysovalantou Nikolaidou, Konstantinos Kouskouras, Nikolaos Fragakis, Vassilios P. Vassilikos, Haralambos Karvounis, Theodoros D. Karamitsos

**Affiliations:** 1Oxford Centre for Clinical Magnetic Resonance Research, University of Oxford, Oxford OX3 9DU, UK; Chrysovalantou.nikolaidou@cardiov.ox.ac.uk; 2First Department of Cardiology, AHEPA Hospital, School of Medicine, Faculty of Health Sciences, Aristotle University of Thessaloniki, 546 36 Thessaloniki, Greece; hkarvounis@auth.gr; 3Department of Radiology, AHEPA Hospital, School of Medicine, Faculty of Health Sciences, Aristotle University of Thessaloniki, 546 36 Thessaloniki, Greece; coskou@auth.gr; 4Third Department of Cardiology, Hippokration Hospital, School of Medicine, Faculty of Health Sciences, Aristotle University of Thessaloniki, 546 43 Thessaloniki, Greece; nfrag@auth.gr (N.F.); vvassil@auth.gr (V.P.V.)

**Keywords:** cardiac magnetic resonance, premature ventricular contractions, ventricular arrhythmia, procainamide, CMR image quality

## Abstract

Acquiring high-quality cardiac magnetic resonance (CMR) images in patients with frequent ventricular arrhythmias remains a challenge. We examined the safety and efficacy of procainamide when administered on the scanner table prior to CMR scanning to suppress ventricular ectopy and acquire high-quality images. Fifty consecutive patients (age 53.0 [42.0–58.0]; 52% female, left ventricular ejection fraction 55 ± 9%) were scanned in a 1.5 T scanner using a standard cardiac protocol. Procainamide was administered at intermittent intravenous bolus doses of 50 mg every minute until suppression of the ectopics or a maximum dose of 10 mg/kg. The average dose of procainamide was 567 ± 197 mg. Procainamide successfully suppressed premature ventricular contractions (PVCs) in 82% of patients, resulting in high-quality images. The baseline blood pressure (BP) was mildly reduced (mean change systolic BP −12 ± 9 mmHg; diastolic BP −4 ± 9 mmHg), while the baseline heart rate (HR) remained relatively unchanged (mean HR change −1 ± 6 bpm). None of the patients developed proarrhythmic changes. Bolus intravenous administration of procainamide prior to CMR scanning is a safe and effective alternative approach for suppressing PVCs and acquiring high-quality images in patients with frequent PVCs and normal or only mildly reduced systolic function.

## 1. Introduction

Cardiac magnetic resonance (CMR) provides excellent assessment of cardiac morphology and function and enables detailed myocardial tissue characterization with a high degree of precision [1]. Using the late gadolinium enhancement technique for regional myocardial scar detection as well as novel T1 mapping techniques for diffuse fibrosis assessment, CMR is widely regarded as the gold standard for identifying structural arrhythmogenic substrate in patients with ventricular arrhythmias, which is often not detected in conventional investigations [2,3]. More importantly, the presence, extent and pattern of myocardial scarring, together with the presumed location of the ventricular arrhythmia, can predict future adverse cardiac events [4,5]. However, when frequent premature ventricular contractions (PVCs) are present during the scan, the quality of the CMR images is severely degraded due to significant R-R interval variations [6]. To overcome this, most scanners offer arrhythmia sorting capabilities, with a prespecified acquisition window allowing only some R-R variation. However, this method results in prolonged acquisition and breath-holding times. Furthermore, it frequently fails when ectopic beats are present in the form of bigeminy or trigeminy, making prospective gating with compromised image quality the only viable option.

An alternative approach to improve image quality when scanning patients with frequent PVCs is the administration of antiarrhythmic medication prior to the scan. The ideal antiarrhythmic for this purpose should have a quick onset but a relatively short duration of action, allowing intravenous (i.v.) administration on the scanner table without the need for further monitoring. Procainamide is a class Ia antiarrhythmic agent which has such features.

We examined the safety and efficacy of procainamide when administered on the scanner table prior to CMR scanning for suppressing ventricular ectopy and acquiring high quality images.

## 2. Materials and Methods

### 2.1. Study Design and Patient Population

The study included 50 consecutive adult patients (median age 53 [42.0–58.0] years; 52% female) referred for a CMR scan due to frequent PVCs or non-sustained ventricular tachycardia on an electrocardiogram (ECG) or a 24 h Holter monitor. The inclusion criteria were a high PVC burden (bigeminy or trigeminy) during CMR scanning that significantly affected image quality and no prior history of heart failure. The exclusion criteria were an estimated glomerular filtration rate <30 mL/min/1.73 m^2^, an allergy to gadolinium-based contrast agents, any contraindication to the MR environment (e.g., MR-unsafe implants or devices, shrapnel injury), pregnancy, claustrophobia, ECG contraindications to procainamide (second- or third-degree heart block, prolonged QT interval, Brugada-type pattern and complete bundle branch block) and patients with a severely reduced ejection fraction on the echocardiogram for safety reasons, as it was not planned for them to stay in the hospital for prolonged hemodynamic monitoring.

### 2.2. CMR Protocol

The CMR studies were performed on a 1.5 T Magnetom Avanto (Siemens, Erlangen, Germany) scanner using phased-array radiofrequency receiver surface coils and dedicated cardiac software. The images were obtained with breath-holding instructions and ECG gating. A standard cardiac protocol was used, including balanced steady state free precession cine images (repetition time 45.30 ms, echo time 1.27 ms, flip angle 55°, field of view 360–420 mm, base resolution 256 × 200, sequential 7 mm slices with a 3 mm interslice gap and 25–30 phases per cardiac cycle), T2-weighted fat-suppressed turbo spin echo imaging (three short-axis 10 mm slices, repetition time of 2 RR intervals, echo time 50 ms, inversion time 170 ms, field of view 360–400 mm and base resolution 192 × 256) and late gadolinium enhancement (LGE) imaging (gadobutrol 0.1 mmol/kg or gadoterate meglumine 0.2 mmol/kg, repetition time 700 ms, echo time 1.37 ms, flip angle 55°, field of view 380–420 mm, base resolution 256 × 232 and sequential 7 mm slices with a 3 mm interslice gap). An operator graded the image quality of the cine images before and after procainamide administration. A 1–4 scale system was used: 4 indicated no artifacts and excellent image quality; 3 indicated minor artifacts with overall good image quality; 2 indicated moderate artifacts significantly affecting image quality and the accuracy of the analysis; and 1 indicated major artifacts, with the scan not able to be analyzed at all.

### 2.3. Procainamide Administration

Procainamide hydrochloride (Biocoryl^®^, vial 1 g/10 mL) was administered on the scanner table at intermittent i.v. bolus doses of 50 mg every minute until the suppression of PVCs was achieved or a maximum dose of 10 mg/kg was reached (Figure 1). Procainamide’s onset of action is 10–30 min after i.v. administration, with peak concentrations achieved in 15–60 min, and a half-life of 2.5–5 h in patients with normal renal function. For bolus i.v. administration, the loading dose was 10–17 mg/kg at a rate of 20–50 mg/min up to a maximum total dose of 1 g. This dose usually achieves therapeutic plasma concentrations ranging from 4–10 mg/L. Toxicity from an i.v. overdose is rare in a monitored clinical setting [7,8]. The blood pressure was measured every minute, and there was continuous monitoring of the heart rate and ECG trace using a magnetic resonance compatible vital signs monitoring system. A 12-lead ECG was performed at the end of drug administration and before patient discharge, 90 min after the scan (approximately two hours after procainamide administration). An emergency cart with resuscitative medications and equipment, such as an external cardioverter defibrillator with external pacing capabilities in case of life-threatening arrhythmias during procainamide administration, were available at an established location outside the scanner room.

### 2.4. Statistical Analysis

Statistical analyses were performed using SPSS statistics software, package version 27.0 (IBM, Armonk, NY, USA) for Windows. We present continuous variables as means (SDs) or medians (25th to 75th percentile), as appropriate. Normality of distribution was tested using the Shapiro-Wilk test. Categorical variables are presented as percentages. Paired t-tests were used to compare the blood pressure and heart rate response of the patients, as well as the image quality before and after procainamide administration. All tests were two-sided, and α was set at 0.05.

## 3. Results

Table 1 summarizes the patient characteristics and CMR findings. The majority of patients (88%) were referred due to frequent PVCs, while 12% also had evidence of non-sustained ventricular tachycardia (NSVT). Most of the patients (84%) had normal findings on echocardiography and no previous medical history, while only a minority had previous coronary artery intervention (4%) and cardiac surgery (2%). One patient had previous chemotherapy for breast cancer, and four patients (8%) had mild or moderately impaired left ventricular (LV) systolic function on an echocardiogram. The CMR scan had normal findings in 42% of patients, while 26% were diagnosed with non-ischemic cardiomyopathy. In 16%, the most likely diagnosis was PVC-related cardiomyopathy, 14% had previous myocarditis, and one patient had dual pathology (dilated cardiomyopathy and previous myocardial infarction).

The majority of the patients (73%) were being treated for their ventricular arrhythmia with oral b-blockers, 7% were on flecainide (Class Ic antiarrhythmic medication, according to the Vaughan-Williams classification scheme) and another 7% were receiving amiodarone, while 13% were not prescribed any treatment by the referring physician. The average dose of procainamide administered intravenously on the scanner table prior to CMR scanning was 567 ± 197 mg (range of 200–1000 mg) with an average duration of administration of 11 ± 4 min. Procainamide successfully suppressed PVCs in 82% of patients (20 patients with complete suppression and 21 with significant reduction (i.e., less than 1 PVC in 10 normal sinus beats), enabling the use of arrhythmia detection and sorting algorithms) resulting in high-quality cine imaging when compared with the initial image (Figure 2, Appendix A). In seven patients, PVCs were only minimally suppressed, and there was no effect by procainamide in two patients. The overall image quality significantly improved after procainamide administration (Table 2).

None of the patients developed significant QRS or QT interval prolongation, advanced atrioventricular block or exacerbation of the ventricular arrhythmia. There was a small but statistically significant drop in blood pressure (BP) after procainamide administration (mean change systolic BP −12 ± 9 mmHg, *p* < 0.001; diastolic BP −4 ± 9 mmHg, *p* = 0.012), but none of the patients developed symptomatic hypotension. There was no significant change in heart rate (HR) (mean HR change −1 ± 6 bpm, *p* = 0.24) (Table 2).

## 4. Discussion

CMR in patients with irregular heart rates, including frequent PVCs, remains challenging due to artifacts. Our study showed that i.v. bolus procainamide successfully suppressed PVCs in 82% of patients with ventricular arrhythmias and significantly improved the image quality, enabling accurate measurements. Procainamide administration on the scanner table was not only effective, but also safe in our group of patients with normal or mildly reduced systolic function.

CMR imaging in patients with arrhythmias constitutes a valuable diagnostic and prognostic tool. More specifically, CMR provides information regarding the underlying cause of and accurately assesses ventricular systolic function, which has been shown to predict adverse cardiac events. The pattern, location and extent of scarring or fibrosis can predict the improvement of cardiac function with treatment as well as future arrhythmic risk [9], in addition to guiding the device treatment or ablation of complex arrhythmias [10]. Importantly, in patients with ventricular arrhythmias, CMR has additional diagnostic value to conventional diagnostic investigations and can identify structural heart disease, even in patients with normal echocardiograms [11].

Cardiac motion during scanning constitutes a major source of image degradation and artifacts in CMR imaging. The problem is solved by ECG-gated image acquisition throughout the cardiac cycle, divided into different phases, and then image reconstruction from the information acquired over several heart beats. When the heart rate is irregular, beat-to-beat variations of the R-R interval cause artifacts during image reconstruction, resulting in significantly impaired image quality [12,13]. In patients with occasional PVCs, arrhythmia rejection algorithms can be used to acquire good quality cine images at the expense of longer breath-holding times. However, in cases of frequent PVCs, arrhythmia sorting is not practical, and the only option is to either revert to triggered data acquisition, accepting the compromised image quality, or use low temporal and spatial resolution real-time imaging [6,12]. Newer techniques, such as accelerated real-time cine imaging with compressed sensing and real-time temporal parallel acquisition, have an acceptable but still inferior diagnostic cine image quality to the standard retrospective ECG-gated breath-holding cine imaging [14,15]. Motion-corrected free breathing averaged late gadolinium enhancement (LGE) imaging provides high quality LGE images in patients with arrhythmias and difficulties with breath-holding [16,17], but this technique is not available to all centers.

Another approach to improve CMR image quality is the administration of an antiarrhythmic medication prior to the scan. We selected i.v. procainamide as an antiarrhythmic agent due its favorable profile in the treatment of ventricular arrhythmias in the acute setting. Procainamide is a class Ia antiarrhythmic agent, whose primary mechanism of action is through inhibition of the rapid inward sodium current, INa. It acts by delaying repolarization and increasing the effective refractory period of atrial and ventricular fibers. The conduction velocity and excitability are also decreased progressively as the drug concentration increases [18]. The ability to suppress automaticity, and the quick onset and short duration of procainamide when administered in bolus i.v. doses, makes it an ideal antiarrhythmic agent for PVC suppression, particularly in patients with normal or only mildly impaired left ventricular (LV) function [19]. Compared to i.v. amiodarone and lidocaine, procainamide is associated with a higher proportion of tachycardia termination and fewer major adverse cardiac events [20,21]. Class Ic antiarrhythmic medications, such as flecainide or propafenone, are also effective in terminating a ventricular arrhythmia, but they were not preferred over procainamide because of potential pro-arrhythmic effects, especially in patients with coronary artery disease or other significant structural heart disease [22,23]. Procainamide’s broad anti-arrhythmic spectrum, including the ability to prolong the effective refractory period together with its potassium channel blocking effects and the quick onset of action, can possibly explain its effectiveness when administered intravenously prior to CMR scanning in suppressing PVCs resistant to the oral antiarrhythmic medications.

Possible acute adverse reactions to procainamide include heart block, bradycardia, tachyarrhythmias or hypotension. Moreover, in patients with severely reduced systolic function, procainamide should be administered with caution because it may produce negative inotropic effects [24]. In the present study, although there was a drop in blood pressure, none of the patients developed symptomatic hypotension; the heart rate was relatively stable, and no patient developed significant ECG changes (i.e., increase of QRS duration >25%), heart block or new or exacerbated ventricular arrhythmia.

Some limitations of this study should be acknowledged. The study group was relatively small, and further studies are needed to assess safety and efficacy in more diseased cohorts of patients (i.e., those with severely impaired LV systolic function). This is important as PVCs are common in patients with heart failure and reduced ejection fraction [25]. Only 8% of patients included in this study had mild or moderately impaired LV systolic function on echocardiography. Finally, in our study, there is no comparison of image quality with the newer imaging techniques described above, which try to mitigate the effects of respiratory motion and arrhythmia [14,15,16,17], as these were not available to our center.

## 5. Conclusions

We propose an alternative approach for high-quality CMR imaging in patients with frequent ventricular arrhythmias and normal or only mildly impaired LV systolic function (i.e., the i.v. administration of procainamide prior to CMR scanning), which appears to be both effective and safe for PVC suppression in this group of patients.

## Figures and Tables

**Figure 1 diagnostics-11-00178-f001:**
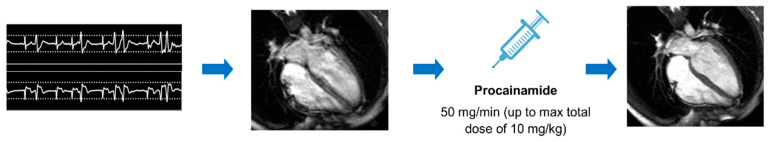
Study protocol. In patients presenting for the Cardiac Magnetic Resonance study with frequent ventricular ectopics and significantly impaired image quality, procainamide was administered on the scanner bed at intermittent intravenous bolus doses. The result was the successful suppression of premature ventricular contractions in most of the patients and a high image quality.

**Figure 2 diagnostics-11-00178-f002:**
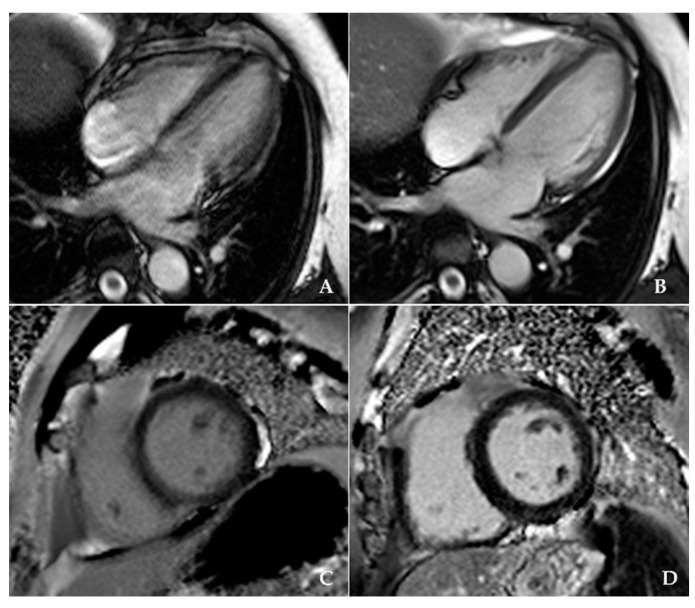
Severely impaired image quality on horizontal long-axis cardiac magnetic resonance cine imaging in a patient with a high ectopic burden (**A**), in contrast to the high-quality images after premature ventricular contraction (PVC) suppression with intravenous administration of procainamide (**B**). Late gadolinium enhancement image with impaired image quality of a patient with frequent PVCs (**C**), compared with the high-quality imaging in another patient with suppression of PVCs with procainamide (**D**).

**Table 1 diagnostics-11-00178-t001:** Patient characteristics and cardiac magnetic resonance (CMR) findings.

**Clinical Characteristics**	
Patients (n)	50
Female, n (%)	26 (52%)
Age (years)	53.0 [42.0–58.0]
Body Surface Area (BSA), m^2^	1.9 ± 0.2
Clinical indication	
Premature ventricular contractions (PVCs), n (%)	44 (88%)
Non-sustained ventricular tachycardia (NSVT), n (%)	6 (12%)
**CMR Measurements**	
LV end diastolic volume (mL)	175 ± 58
LV end systolic volume (mL)	83 ± 49
LV ejection fraction (%)	55 ± 9
RV end diastolic volume (mL)	154 ± 42
RV end systolic volume (mL)	64 ± 31
RV ejection fraction (%)	60 ± 8
**CMR Diagnosis**	
Normal, n (%)	21 (42%)
Non-ischemic cardiomyopathy, n (%)	13 (26%)
PVC-related cardiomyopathy, n (%)	8 (16%)
Previous myocarditis, n (%)	7 (14%)
Dual pathology (DCM and MI), n (%)	1 (2%)

Values are shown as n (%), mean ± standard deviation if they have normal distribution, while values without normal distribution are shown as median with interquartile range. CMR = cardiac magnetic resonance; DCM = dilated cardiomyopathy; LV = left ventricular; MI = myocardial infarction; RV = right ventricular; and PCI = percutaneous coronary intervention.

**Table 2 diagnostics-11-00178-t002:** Procainamide dose and effect on hemodynamic parameters and arrhythmia burden.

**Procainamide Administration**	**Before**	**After**	***p*-Value**
Mean procainamide dose (mg): 567 ± 197 Range (mg): 200–1000	-	-	-
Systolic blood pressure (mmHg)	133 ± 19	121 ± 17	<0.001
Diastolic blood pressure (mmHg)	68 ± 14	64 ± 12	0.012
Heart rate (bpm)	75 ± 12	74 ± 13	0.24
Image Quality (1 bad, 2 moderate, 3 very good, 4 excellent)	1.62 ± 0.49	3.46 ± 0.51	<0.001
**Result**	**No. of Patients (%)**		
Complete PVCs suppression	20 (40%)		
Significant PVCs reduction	21 (42%)		
Minimal PVCs suppression	7 (14%)		
No response	2 (4%)		

Values are shown as mean ± SD or n (%). PVCs = premature ventricular contractions.

## Data Availability

The data presented in this study are available on request from the corresponding author. All the data are stored at the local research server and are not publicly available.

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
