# Peer review of "Bolus Intravenous Procainamide in Patients with Frequent Ventricular Ectopics during Cardiac Magnetic Resonance Scanning: A Way to Ensure High Quality Imaging"

_diagnostics, 2021, doi:10.3390/diagnostics11020178_

Round 1

Reviewer 1 Report

The authors present results on injection of procainamide to reduce arrhythmia in PVC patients during CMR scans. Safety and efficacy are evaluated. The work represents a potential method to improve cardiac MR image quality in hospital settings where state-of-the-art MRI techniques for cardiac MR for artifact reduction and/or image quality enhancement are not available. 

Overall ambition of the study, results, discussion of the generated data as well as limitations of the study are well presented. In order to further improve the manuscript's quality and qualify for publication in this journal, a few minor recommendation shall be implemented.

Line 85:  which product containing procainamide has actually been used? Please specify.

Line 90:  how long have the patients been observed after injection of procainamide before discharge?

Line 101:  which statistical analysis has been performed on which data? Please specify the tests utilized.

Results section: 

The manuscript would benefit from a tabular presentation of the safety results in the main text. Please include a table summarizing the described changes in cardiac parameters. Further side effects have been reported for procainamide; have any of those been observed? Please comment. 

A quantitative assessment of the observed improvements of CMR image quality is missing. The manuscript would strongly benefit from a quantitative analysis of the image quality improvement, especially as image quality has not directly been compared to other available techniques. Please also comment on how the observed improvement in image quality influenced the diagnosis in the patients enrolled. 

Author Response

Point-by-point response to comments:

The authors present results on injection of procainamide to reduce arrhythmia in PVC patients during CMR scans. Safety and efficacy are evaluated. The work represents a potential method to improve cardiac MR image quality in hospital settings where state-of-the-art MRI techniques for cardiac MR for artifact reduction and/or image quality enhancement are not available. 

Overall ambition of the study, results, discussion of the generated data as well as limitations of the study are well presented. In order to further improve the manuscript's quality and qualify for publication in this journal, a few minor recommendations shall be implemented.

Thank you for your constructive comments which have been very helpful in improving our manuscript. We have incorporated nearly all of your suggestions and we believe that your comments have significantly improved the manuscript. We have uploaded a revised manuscript with changes highlighted in yellow.

Line 85:  which product containing procainamide has actually been used? Please specify.

The name of the product containing procainamide has been added.

Line 92: Biocoryl®, vial 1g/10ml

Line 90:  how long have the patients been observed after injection of procainamide before discharge?

The time of observation of patients after injection of procainamide before discharge has been included.

Lines 102-104: A 12 lead ECG was performed at the end of drug administration and before patient discharge, 90 minutes after the scan (approximately two hours after procainamide administration.

Line 101:  which statistical analysis has been performed on which data? Please specify the tests utilized.

As described, continuous variables (such as demographic parameters, CMR measurements, blood pressure and heart rate) are presented using mean value ± standard deviation and categorical parameters as percentages. We compared continuous variables of the patients before and after procainamide administration with the paired samples t-test (Lines 114-118).

Results section: 

The manuscript would benefit from a tabular presentation of the safety results in the main text. Please include a table summarizing the described changes in cardiac parameters. Further side effects have been reported for procainamide; have any of those been observed? Please comment. 

A quantitative assessment of the observed improvements of CMR image quality is missing. The manuscript would strongly benefit from a quantitative analysis of the image quality improvement, especially as image quality has not directly been compared to other available techniques. Please also comment on how the observed improvement in image quality influenced the diagnosis in the patients enrolled. 

Table S1 with the safety results has now been included in the main text as Table 2. Procainamide effect on arrhythmia burden is now also presented in the table. None of the patients developed any other side effects from procainamide, such as significant QRS or QT interval prolongation, advanced atrioventricular block or exacerbation of the ventricular arrhythmia (Lines 156, 157).

Thank you for your suggestion to perform a quantitative assessment of the image quality. We used an operator graded image quality scale (scales 1 to 4) to score image quality (Lines 85-90). This showed that there is a statistically significant improvement in image quality after procainamide administration (Lines 147-148, Table 2).

We also added more literature on CMR and cardiac arrhythmias and the technical challenges in patients with arrhythmia in the Discussion (lines 165-185), to show that diagnosis is difficult or impossible when image quality is impaired. 

Reviewer 2 Report

The Authors presented an article "Bolus Intravenous Procainamide in Patients with Frequent 2 Ventricular Ectopics during Cardiac Magnetic Resonance Scanning: A Way to Ensure High Quality Imaging".

Specific comments:

1) As you mentioned you have a limitation in procainamide safety experiments. So I think much more discussion and information relative to a dose of the drug and Its safety/toxicity should be presented. Some works about its safety in a dose-dependent manner should be presented and cited. 

2) The obtained results and the drug safety according to the procainamide discussion should be inserted. 

"The average dose of procainamide administered on the scanner table prior to CMR scanning was 567 ± 197 124 mg (range 200-1000 mg)" The dose of 567 I think can be safe, but 2000 mg may be very high and dangerous for the patient. It is possible to obtain good results using a low dose of the drug? I think the authors should present some information or discussion about the dose (the range 200-1000 mg is very high) and other factors and obtained results.

3) I think, It can be mentioned a bit more literature works/reviews according to the presented problem. Maybe some more works of professor Karamitsos or others.

4) Fig. 1 will looks better If it will be in a horizontal way. Table 1 will read better if it will be without so many abbreviations. Maybe It can be done in the manner as a "full name (abbreviation)" in the column. For example, body surface area (BSA).

Author Response

Thank you for your constructive comments which have been very helpful in improving our manuscript. We have incorporated nearly all of your suggestions and we believe that your comments have significantly improved the manuscript. We have uploaded a revised manuscript with changes highlighted in yellow.

Point-by-point response to comments:

1) As you mentioned you have a limitation in procainamide safety experiments. So I think much more discussion and information relative to a dose of the drug and its safety/toxicity should be presented. Some works about its safety in a dose-dependent manner should be presented and cited. 

and

2) The obtained results and the drug safety according to the procainamide discussion should be inserted. 

Thank you for commenting on the doses and safety of procainamide administered to the patients. The doses administered were from 200 mg to 1000 mg (Line 141) which is the maximum suggested safe dose; we did not administer 2000 mg to any patient. Based on your comment, in order to prove that the doses we administered were safe we added to the Methods section (lines 94-100) information about dose, pharmacokinetics and safety. Also, we included in the main text as Table 2 the safety results shown previously in Table S1.

3) I think, it can be mentioned a bit more literature works/reviews according to the presented problem. Maybe some more works of professor Karamitsos or others.

More literature on CMR and cardiac arrhythmias and the technical challenges in patients with arrhythmia have been added to the Discussion (lines 165-186).

4) Fig. 1 will looks better if it will be in a horizontal way. Table 1 will read better if it will be without so many abbreviations. Maybe it can be done in the manner as a "full name (abbreviation)" in the column. For example, body surface area (BSA).

Figure 1 is now in horizontal orientation.

    Most of the abbreviations have been removed from Table 1. We agree that it is easier to read now.
